# Experimental investigation of the effects of floating wind turbine motion on a downstream turbine performance and loads

Alessandro Fontanella<sup>1</sup>, Stefano Cioni<sup>2</sup>, Francesco Papi<sup>2</sup>, Sara Muggiasca<sup>1</sup>, Alessandro Bianchini<sup>2</sup>, and Marco Belloli<sup>1</sup>

<sup>1</sup>Department of Mechanical Engineering, Politecnico di Milano, via La Masa 1, 20156 Milano, Italy.

<sup>2</sup>Department of Industrial Engineering, Università degli Studi di Firenze, Via di Santa Marta 3, 50139 Firenze, Italy.

Correspondence: Alessandro Fontanella (alessandro.fontanella@polimi.it)

**Abstract.** This study investigates how the motion of a floating wind turbine affects the aerodynamic performance and dynamic loading of a downstream turbine operating in its wake. Wind tunnel experiments were conducted using a two-turbine setup, where the upstream turbine was subjected to controlled platform motions (both sinusoidal and wave driven) while the downstream turbine remained fixed and was tested in multiple relative positions. Results show that large-amplitude, low-frequency,

- sinusoidal motions of the upstream turbine, especially in crosswind and yaw directions, can increase the power output of the downstream turbine under low-turbulence conditions and at short turbine spacing (3-5 rotor diameters). The largest relative power gain reached 26% over the fixed case, although absolute gains remained moderate. The gains obtained under idealized sinusoidal motions were replicated in cases with realistic-wave-driven motions when wind and waves were aligned, but not when wind–wave misalignment introduced crosswind movements of the upstream wind turbine. In parallel, motion of the up-
- stream turbine increased the dynamic loading on the waked turbine. These effects varied with turbine spacing and alignment, and were more pronounced in sinusoidal motion cases. Still, similar mechanisms were observed under wave-induced motion. Overall, these findings underscore that platform-induced wake dynamics are not a secondary effect, but a key driver that must be considered in the design and operation of floating wind farms.

# 1 Introduction

- The deployment of floating offshore wind farms is a key strategy to harness wind energy in deep-water regions where fixedbottom turbines are not feasible. While the floating wind technology enables access to abundant yet untapped wind resources, it also introduces additional complexities, especially regarding system dynamics due to the large motions of platforms and their influence on the wind turbine aerodynamic response. In particular, the aerodynamic interactions between wind turbines through wakes, which is already a critical factor in bottom-fixed wind farms, become even more complex in floating wind
- farms, as the aforementioned large platform motions typical of this technology may lead to more unsteady and irregular wake flows. These wake effects can lead to substantial power losses, increased fatigue loads, and control challenges at the turbine and plant levels. Despite their importance they remain poorly understood in the context of floating wind farms (Meyers et al., 2022) and one reason for the limited understanding of aerodynamic interactions in floating wind farms is the lack of wake and

performance data from operational plants. To date, only a few floating wind farms have been constructed, each comprising a small number of turbines (Chitteth Ramachandran et al., 2022).

Most available studies focus on the wake behavior of individual floating turbines and do not address how turbines interact through wakes within a full wind farm. A recent study by Angelou et al. (2023) examined the far wake of a 6 MW floating wind turbine in the Hywind Scotland offshore wind farm, the world's first floating wind farm, using LiDAR measurements. Its findings indicate that the wakes of floating turbines, for small-amplitude dynamic motions of the platform, exhibit time-averaged

characteristics similar to those of fixed-bottom turbines. Under these conditions, atmospheric turbulence was identified as the dominant mechanism for wake recovery, with platform motion having a negligible effect. However, the investigation focused on conditions in which the turbine operated outside the wake of upstream units, and therefore did not address the impact of upstream wakes on turbine performance within the array.

The impact of large platform motions on the wake behavior of floating wind turbines has been instead extensively investigated through wind tunnel experiments, which offer controlled and repeatable conditions ideal for isolating aerodynamic phenomena. Most of these studies have employed model-scale wind turbines subjected to unidirectional sinusoidal motions of large amplitude and low frequency, typically in low-turbulence inflow.

The experiment of Bayati et al. (2017a) analyzed the effect of surge motion on the near wake of a 1:75 scale model of the DTU 10 MW wind turbine (Bak et al., 2013), demonstrating that platform motion introduces wind speed fluctuations in the

- wake. Building on this, Fontanella et al. (2022b) explored additional surge motion conditions and showed how motion affects the formation and evolution of tip vortices near the rotor. In a later study, Fontanella et al. (2022a) investigated the wake of a 1:100 scale model of the IEA 15 MW reference turbine (Gaertner et al., 2020) under platform motions in along-wind and crosswind directions. The results indicated that all motion types induced velocity fluctuations in the wake, although mean velocity in the near wake was slightly reduced compared to the fixed-turbine case. Messmer et al. (2024a) further examined wake
- behavior under harmonic surge and sway motions in laminar inflow conditions. The study found that sway motion enhances lateral meandering, while surge motion induces wake pulsation; both contribute to improved mixing and recovery of the far wake. In a follow-up study, Messmer et al. (2024b) assessed the effect of surge motion under inflow turbulence intensities up to 3%, showing that increased turbulence significantly diminishes the influence of platform motion on the far wake recovery. Bossuyt et al. (2023) investigated a 1:400 scale floating wind farm with twelve turbines in a wind–water tunnel under a sheared
- inflow with 10% turbulence intensity. Despite the challenges of simultaneously scaling aerodynamics and hydrodynamics, the study revealed clear differences in wake recovery linked to the periodic rotor motions. Wave-induced yaw and pitch led to synchronized vertical wake oscillations that enhanced recovery but also increased turbulence levels, potentially affecting the unsteady loading on downstream turbines.

Taken together, these experiments consistently demonstrate that large rotor motions influence wake development, generating flow structures with the same periodicity as the platform motion. These dynamic features, primarily seen as pulsations in velocity and increased lateral meandering, are absent in wakes of bottom-fixed turbines. Under nearly laminar conditions, such wake dynamics can enhance recovery, but their effect becomes negligible as inflow turbulence increases. At low but

realistic turbulence intensities, platform motions still generate coherent periodic structures, but they do not lead to substantial improvements in wake recovery.

- The limited availability of experimental data has led to diverse modeling approaches for floating wind farms, often yielding scattered results. Carmo et al. (2024) used FAST.Farm, which neglects dynamic wake effects due to platform motion, modeling only the wake vertical deflection due to average rotor inclination. This simplification was deemed acceptable under typical turbulence conditions, where wake pulsing effects are expected to be minimal. In contrast, Ramos-García et al. (2022) used a vortex-multibody solver to show that platform motions in an upstream turbine enhance wake breakdown, increasing power and
- thrust in downstream turbines and that this effect diminishes with higher inflow turbulence. Simulations with prescribed surge and pitch motions confirmed that platform dynamics significantly affect wake behavior and downstream loading, especially deeper in the array where resonant interactions may occur.

A number of numerical studies using computational fluid dynamics (CFD) have investigated wake behavior in floating wind turbines, often focusing on isolated machines (Micallef and Rezaeiha, 2021). Moving toward farm-scale interactions,

- Arabgolarcheh et al. (2023) used unsteady Reynolds-averaged Navier–Stokes (URANS) simulations to examine the effect of prescribed surge motion on two aligned floating wind turbines. While the mean thrust and power output showed little difference between fixed and surging upstream rotors, the motion generated periodic wake structures that increased dynamic loading on the downstream turbine. Similarly, Li et al. (2025) employed large eddy simulation (LES) to study wake interactions between two surging turbines and found that, under laminar inflow, the downstream power increased significantly. However,
- this gain became negligible under more realistic turbulence levels. Additional insight into the physical mechanisms governing this behavior has recently been provided by Pagamonci et al. (2025), who conducted LES-based CFD simulations on the same wind tunnel-scale turbine used in the present study. Their results show that under turbulent inflow conditions, tip vortex breakdown occurs much closer to the rotor compared to laminar inflow, leading to markedly different wake development than what is predicted under idealized flow conditions.
- Although CFD studies provide valuable insights into wake dynamics and turbine interactions, their results often vary due to differences in turbulence modeling, inflow conditions, and domain size. This variability limits their reliability in predicting wake effects in realistic floating farm configurations. This motivates the need for controlled experimental data, which can serve both to understand key physical mechanisms and to validate or calibrate numerical tools.

This study investigates, through wind tunnel experiments, how wakes generated by floating wind turbines affect the dynamic loading and power output of a downstream turbine operating in their wake. Specifically, it seeks to address two key research questions:

- 1. Do the unsteady flow disturbances induced by platform motion propagate downstream and influence the load response of a waked turbine?
- 2. Can a turbine operating in the wake of a moving upstream rotor extract more power than one operating behind a bottom-fixed turbine?

The experiment and its results are presented in the following sections. Section 2 describes the methodology of the study, including the experimental setup, the operating conditions of the wind turbines, and the platform motions. Section 3 presents and analyzes experimental results. In Section 4, these results are discussed with the aim of evaluating their relevance and applicability to the design and operation of floating wind farms. This discussion also considers the limitations and simplifications inherent to the experimental setup. Finally, Section 5 summarizes the main findings of the study and outlines the conclusions.

2 Methodology

The experimental campaign was conducted in the atmospheric-boundary layer test section of Politecnico di Milano wind tunnel, which is 13.84 m wide by 3.84 m high by 35 m long and used two wind turbines (WT) that are 1:75 scale models of the DTU 10 MW reference wind turbine (Bak et al., 2013). In the experiment, the flow speed was scaled by a factor 1:3 compared to a full-scale environment. The time scaling in the experiment was determined by the geometric and velocity scales, resulting in a time scale ratio of 1:25. Consequently, the wind farm response in the experiment occurred 25 times faster than at full scale. In the experiment, only the upstream turbine was subjected to platform motions representative of a floating wind turbine, while the downstream turbine was fixed at the tower base. This approach avoids introducing additional variables that could make it more difficult to identify the specific influence of the upstream turbine dynamics on the downstream turbine response.

Similar methodologies have been employed in previous wind tunnel studies on wake interactions, particularly in active wake control research, where only the most upstream turbine is dynamically actuated, and downstream turbines are operated with fixed settings (Van Der Hoek et al., 2024; Van Der Hoek et al., 2024). Once the effects of upstream rotor motion are well understood, the analysis can be extended to more realistic scenarios that also include the motion of the downstream turbine.

The experiment with two turbines followed the one of Fontanella et al. (2024), that was performed in the same wind tunnel using the same wind turbine and robotic platform and measured the wake of the upstream wind turbine at the downstream locations where WT2 was placed in the current experiment. Hence, the inflow caused by the wake of the upstream turbine on the downstream turbine is precisely known because it has been already characterized.

#### 2.1 Experimental setup

The experimental setup is shown in Fig. 1. The upstream wind turbine (WT1) was mounted on a six-degrees-of-freedom robotic platform that mimics the rigid-body motions of floating foundations. The downstream wind turbine (WT2) was mounted on a rigid support structure connecting the tower base to the wind tunnel floor.

The two wind turbines are identical. Their main geometric parameters are summarized in Table 1. The rotors of the wind turbines have a diameter (D) of 2.38 m and were designed to replicate at small scale the normal load distribution of the DTU 10 MW blade and the resulting thrust force (Bayati et al., 2017b). The shaft of the wind turbines had a tilt of 5°, matching the

120 full-scale version of the DTU 10 MW. During wind tunnel tests, the towers of the two wind turbines were tilted at a negative angle of  $5^{\circ}$  to ensure the rotors were vertical to the wind tunnel floor. This simplification was intended to limit the analysis to axial thrust and torque in scenarios without prescribed platform motion and with the turbine operating in undisturbed flow.