# Peer review of "Experimental investigation of the effects of floating wind turbine motion on a downstream turbine performance and loads"

_Wind Energy Science, 2025_

## Referee Comment (RC1)

**Review of manuscript wes-2025-106 "Experimental investigation of the effects of floating wind turbine motion on a downstream turbine performance and loads" by A. Fontanella, S. Cioni, F. Papi, S. Muggiasca, A. Bianchini, and M. Belloli.**

This manuscript presents the experimental investigations on the effect of the downscaled upstream floating wind turbine's motion on the performance and loading of a downstream fixed turbine. The upstream turbine is subjected to sinusoidal and wave-driven motions in different DOFs in both wind-wave aligned and mis-aligned cases, while the downstream turbine is placed at different streamwise and lateral positions to make it operate under partial or full wake conditions. Crosswind and yaw motion were found to enhance the power capture and dynamic loading of the downstream turbine when placed at the investigated downstream locations of 3D-5D. Motions under wind-wave alignments were found to have significantly higher effect compared to the motions with wind-wave misalignments. The study investigates the important and relevant topic of wake-interactions between the floating offshore wind turbines focusing on power capture and dynamic loadings for different motion conditions.

The rationale for the study with a brief literature review is presented in the first section followed by the section in the methodology, which explains the experimental setup, motion parameters and motion types used in the study. Experimental results on the power and dynamic loading on the downstream turbine are presented in the following section highlighting the effects of different motion DOFs and types. Discussion of the results is performed identifying the need of tradeoffs between the increased power and dynamic loading while designing the floating offshore wind farms. Concluding remarks are provided suggesting future studies under realistic operating conditions for further investigation of wake interactions. The manuscript is well written, organized and easy to follow. The overall experimental study is highly motivated, and results are well documented. Justification of the observed results on power and loadings are mostly provided in reference to the previous experimental results on the wake. However, some discussions on the effect of the induction of the downstream turbine in the wake and its interactions with the waked inflow under different motion conditions of the upstream turbine would make the study more robust. Besides, the following comments should be addressed to improve the quality of manuscript:

1.  Page 7, Line 171: The operating conditions of WT2 were defined in terms of rotor-effective wind speed ($U_{RE}$), at each condition with fixed WT1. Optimal tip-speed ratio was calculated based on $U_{RE}$, which changes depending on the dynamic motion of WT1. How is the optimal-tip-speed ratio maintained for the cases with dynamic motion of WT1 and how does it affect the comparison of the power capture between different cases?

2.  For placement of downstream turbine at 3D1D and 5D1D, in which the rotor speed was set to maintain the same thrust force as in the free-stream condition, how is the optimal tip-speed ratio maintained in the presence of speed-up? This should be clarified in the manuscript.

3.  Is the blockage correction performed? If so, how is it performed when two turbines are present? If the speed ups at 5D1D are attributed to blockage, how does the results obtained at that location can be related to the results in the atmospheric flows where the incoming flow essentially remains unbounded and the blockage is low? It would be nice, if the authors provide clarification.

4.       Page 9, Line 210: The relation expressing the motion of hub in pitch motion $\dot{d}_{hub,x} = \dot{d}.r_{hub}$ appears to be dimensionally inconsistent and needs revision. The parameters $d$ and $a_m$ should be defined separately for pitch motion. Moreover , $r_{hub}$ is has been defined as a distance between the rotor apex and platform rotation point. I think, this should be the distance between the rotor hub and platform rotation point. This should be corrected.

5.       Page 10, Line 243: It would be nice if the range of large amplitudes are presented.

6.       Page 10, Line 244: How are the OpenFAST and SOFTWind System simulations related? Are they performed for the same wind/wave conditions? Some comments for clarifications would be helpful.

7.       Page 17, Line 384: The amplitude of pitch motion appears to be too small (and like that of surge motion) to bring noticeable vertical wake movement. It would be nice to corroborate the presence of second harmonic in WT2 in lieu of the wake measurements performed in the present (or past) campaigns.

8.       Page 19, Line 430: The reduction in net load oscillations appears to be strange. Should the higher oscillations in velocity not increase the fluctuations in load? Or does it increase the fluctuations in side-to-side loads not the thrust load of the turbines? Further, clarifications on the phenomenon to justify the results would improve the quality of the manuscript.